# Older Is Not Necessarily Better: Decolonizing Ifugao History through the Archaeology of the Rice Terraces

Stephen B. Acabado [1,*] and Marlon M. Martin [2]

1   Department of Anthropology, University of California (UCLA), Los Angeles, CA 90095, USA
2   Save the Ifugao Terraces Movement, Kiangan 3604, Ifugao, Philippines; marlon.martin12@yahoo.com
*   Correspondence: acabado@anthro.ucla.edu

**Abstract:** This study examines the intersection of archaeological data and community narratives in interpreting the Ifugao Rice Terraces in the Philippines, a UNESCO World Heritage Site. Long regarded as 2000-year-old symbols of an uncolonized cultural past, recent research challenges this view, suggesting a 16th-century origin coinciding with Spanish contact. The longstanding characterization of the Ifugao Rice Terraces as 2000-year-old monuments cemented a perception of Ifugao culture as static and unchanging, overshadowing the dynamic cultural practices that have persisted and evolved over the centuries. It is crucial to recognize that these terraces are not frozen in time but are active representations of Ifugao's living culture, which has continually adapted to social, environmental, and historical changes while maintaining its distinct identity. This paradigm shift, supported by radiocarbon dating and ethnohistorical analysis, aligns more closely with local oral histories and portrays the Ifugao not as passive inheritors of tradition but as active participants in their history. We argue for the integration of scientific data with community stories, presenting a holistic understanding of the terraces as dynamic elements of Ifugao resilience and identity. The findings advocate a move away from romanticized historical interpretations toward a narrative that respects the complexity and adaptability of Indigenous cultural landscapes.

**Keywords:** decolonizing archaeology; community engagement; Ifugao; Philippines; rice terraces

## 1. Introduction

The discipline of archaeology often finds itself at the crossroads of past and present, where the narratives of history meet the interpretations of the modern era. This intersection is fraught with challenges, particularly the tendency to romanticize the past, leading to a skewed understanding that often intertwines with national consciousness. Such romanticized views are not merely academic indulgences; they shape identities, policies, and perspectives of entire communities. A striking example of this phenomenon is the Ifugao Rice Terraces in the Philippines, a UNESCO World Heritage Site (Figure 1). These terraces are not just remarkable feats of agricultural engineering but also a palimpsest onto which various narratives of history, identity, and culture have been projected (Figure 2).

We underscore the deeply ingrained controversy centered around the ethnohistory of the rice terraces and propose a pathway to resolution that involves collaboration between archaeologists and the Ifugao community. It foregrounds the notion that the traditional archaeological narrative, heavily influenced by early 20th-century anthropologists like Roy Franklin Barton [1] and Henry Otley Beyer [2], has promulgated a view of the Ifugao and their terraces as static relics of a distant past. This perspective, entangled with nationalist pride, asserts the terraces' antiquity as a symbol of uncolonized purity and cultural identity. The terraces are reflective of cultural and historical dynamism of the Ifugao, but the 2000-year old origin argument appears to make these terraces static.

This paper aims to demystify, and in the process, decolonize, archaeological and historical narratives developed in the early 1900s by pioneer anthropologists who worked

in the Cordillera region of the Philippines. In addition, the dating of the Ifugao Rice Terraces has been part of the nationalist discussions that older is better, but at the same time, relegating the Ifugao to the sidelines of history. This can be achieved by actively engaging stakeholder and descendant communities, as exemplified by our work.

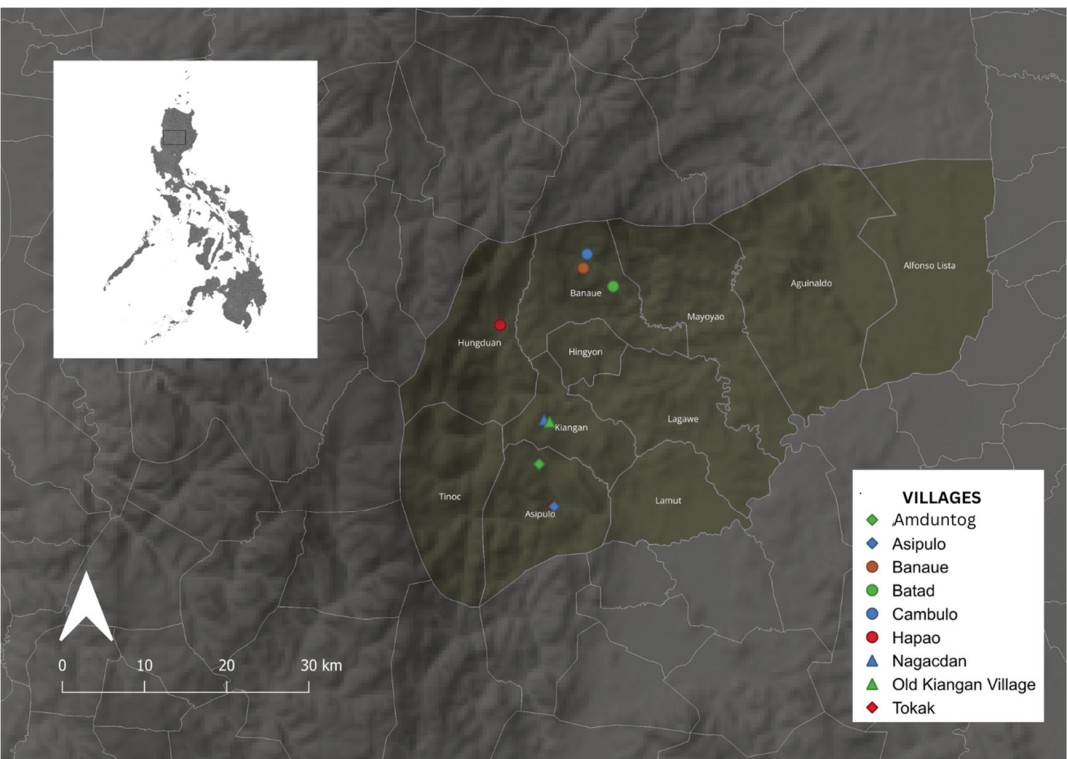

**Figure 1.** Ifugao Province, with approximate locations of major sites mentioned in the text. The municipality of Asipulo includes Tokak and Amduntog; Cambulo is adjacent to Batad; and Old Kiyyangan Village is adjacent to Nagacadan (Map by NJ Roxas and the IAP).

For decades, the dominant narrative, championed by early anthropologists such as Roy Franklin Barton [1] and Henry Otley Beyer [2], posited that the Ifugao Rice Terraces were over 2000 years old. This narrative was more than just a chronological assertion; it was a statement of cultural identity. It presented the Ifugao as custodians of an ancient, unchanging tradition, existing in isolation from the influences and tumults that shaped the rest of the archipelago. In this view, the terraces were not only an agricultural wonder but also a symbol of a pure, uncolonized past. This interpretation has been echoed in academic circles, popular media, and even in the descriptions of the terraces in UNESCO documents, reinforcing the image of the Ifugao as a timeless entity, disconnected from the currents of history.

While only two published materials purports to argue for the 2000-year origin of the terraces, this has become entrenched in the Philippine historical narratives because of the inclusion of the dating on Philippine curricula. As such, our approach challenges this long-standing narrative by actively involving the Ifugao community in the research process. By doing so, we uncover a more nuanced and historically grounded understanding of the Ifugao Rice Terraces and the people who have shaped their history over generations.

Recent archaeological findings have challenged this long-standing belief [3–7]. Utilizing a combination of radiocarbon dating, archaeobotanical analysis, and ethnohistorical research, these studies suggest a much later inception of the terraces, likely around the period of Spanish colonization in the 16th century. This revelation is not just a mere adjustment of dates; it represents a paradigm shift in how the history of the Ifugao and their iconic terraces is understood. It suggests that rather than being a relic of a distant past, the terraces are a dynamic, living testament to the Ifugao's resilience and adaptability in the face of external influences.

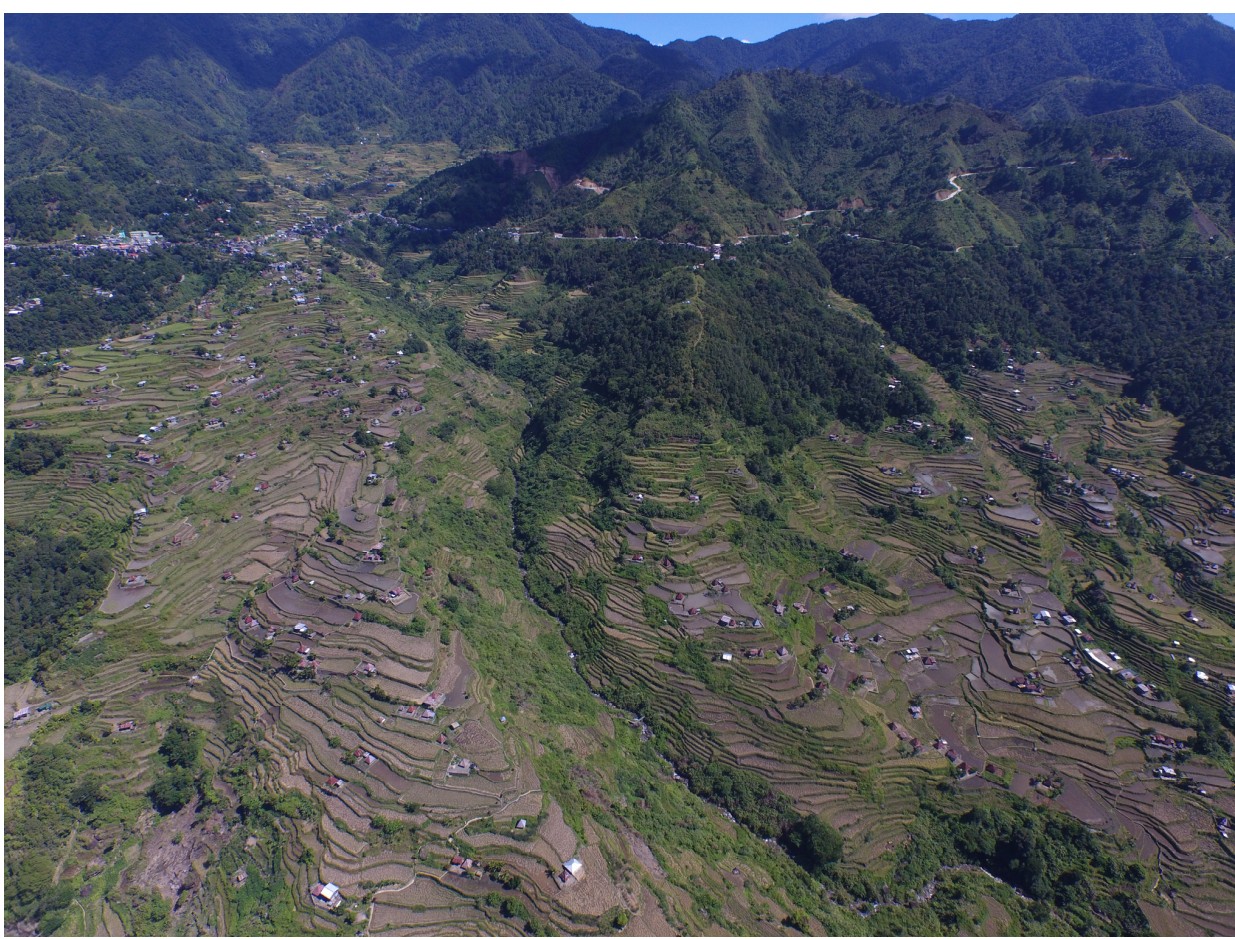

**Figure 2.** The Mayoyao Rice Terraces, one of the five terrace clusters listed on the UNESCO Cultural Landscapes World Heritage Sites (Photo by Bhong Tawana and the IAP).

The Implications of this new interpretation force the rethinking of dominant historical narratives. For one, it challenges the narrative of the Ifugao and their terraces as isolated and unchanging. It paints a picture of a people who are not mere passive recipients of history but active agents who have continuously adapted and evolved in response to changing circumstances. This view aligns more closely with the oral histories and narratives of the Ifugao community, which have often been sidelined in academic discourse in favor of more 'scientific' interpretations.

Community stories and oral histories are invaluable in this recontextualization, providing a complementary, as well as supplementary, narrative to the archaeological data. In Ifugao, these oral traditions serve as living records chronicling the collective memory and societal changes over generations. They provide insight into the cultural significance of the terraces, detailing the struggles and achievements of the Ifugao people that are not always visible in the archaeological record. These narratives often highlight the terraces as symbols of identity and self-determination, countering the narratives of isolation and stasis with tales of dynamism, community cohesion, and resistance to colonial pressures.

These oral traditions offer a nuanced understanding that can challenge and refine archaeological interpretations. They can affirm dates and events, suggest alternative readings of material culture, and illuminate the sociopolitical structures that facilitated the construction and maintenance of the terraces. For example, the community's recounting of the forced relocations and the strategic establishment of settlements away from Spanish incursions provides context to the changes in settlement patterns inferred from archaeological digs. Moreover, the oral histories of agricultural practices, rituals, and social organization offer a vivid picture of the day-to-day life that shaped the terraces' landscape. When oral

histories are integrated into the archaeological discourse, they enrich the narrative, allowing a more holistic reconstruction of the past that acknowledges the active role of the Ifugao in shaping their history and landscape.

Our archaeological work in Ifugao represents a movement towards the decolonization of archaeology. In our work, we define decolonization as the process of re-evaluating and transforming practices to remove colonial biases and structures. It seeks to decenter Western epistemologies, prioritize Indigenous perspectives, and value local knowledge systems on par with scientific methods. This means recognizing and rectifying the historical power imbalances in the interpretation, narration, and stewardship of cultural heritage i.e., [8–10]. Decolonizing archaeology entails actively involving descendant communities in the research process, respecting their oral traditions, and acknowledging their sovereignty over their cultural narratives and material past [11–13].

### 1.1. Changing the Narrative

The 2000-year dating of the Ifugao Rice Terraces has played a pivotal role in shaping perceptions of Ifugao culture as static and unchanging through millennia. The assertion of such antiquity has been intertwined with the narrative of an unchanging culture, one that has ostensibly remained isolated and untouched by the broader historical and cultural dynamics that have swept through the rest of the Philippines and Southeast Asia.

The 2000-year-old construction dating assumes static culture since archaeological models strongly argue that if intensive agriculture has been documented, then we expect to see developments in other aspects of culture. It is then assumed that the Ifugao did not do anything else but plant rice for two millennia. This perspective also suggests that the Ifugao people have managed to preserve their way of life, with little to no external influence, maintaining the same techniques of rice cultivation and terrace construction passed down through generations. It romanticizes the Ifugao as custodians of an ancient tradition, living relics of a precolonial past. This narrative has been appealing because it offers a sense of timelessness and purity, resonating with nationalist sentiments that valorize Indigenous cultures as symbols of national identity and resistance to colonial forces.

#### 1.1.1. The Notion of Isolation

Closely related to the idea of an unchanging culture is the notion of isolation. The dating implies that the Ifugao Rice Terraces—and by extension, the Ifugao people—have existed in a bubble of isolation, shielded from the transformative influences of trade, colonialism, and cultural exchange that have characterized the history of the region. This perception of isolation has been used to support the argument that the terraces are a testament to Indigenous ingenuity and resilience, constructed and maintained without external influence.

However, this view oversimplifies the complex historical and cultural dynamics that have shaped the Ifugao region. It overlooks evidence of trade, interaction, and conflict with neighboring groups and colonial powers, which have undoubtedly influenced Ifugao society. Moreover, it disregards the adaptability and agency of the Ifugao people, who have navigated the challenges posed by external pressures and internal developments over centuries.

#### 1.1.2. Critique and Reevaluation

Recent archaeological findings and ethnohistorical research challenge the narrative of an unchanging, isolated culture. Studies suggesting a much later inception of the terraces, around the time of Spanish colonization in the 16th century, highlight the dynamic nature of Ifugao society. This evidence indicates that the terraces are not relics of a static past but rather the outcome of a society's adaptive strategies in response to social, environmental, and political pressures.

The reconsideration of the terraces' dating prompts a broader reflection in how cultures are perceived and represented in historical narratives. It underscores the need to

move beyond simplistic notions of timelessness and isolation, recognizing the fluidity and interconnectedness of human societies. Acknowledging the terraces as a product of historical contingencies rather than a symbol of unchanging tradition allows for a more nuanced understanding of the Ifugao's cultural resilience and ingenuity.

Emerging archaeological and ethnohistorical evidence challenges notions of an unchanging and isolated culture, highlighting the dynamic and interconnected nature of Ifugao history. This reevaluation not only enriches our understanding of the Ifugao but also contributes to broader discussions about the representation and interpretation of Indigenous cultures in historical narratives.

*1.2. Decolonizing Archaeology*

In the context of the Ifugao Rice Terraces, the process of decolonizing archaeology entails a shift away from a Western-centric interpretation that idealizes and freezes the Ifugao people in a fixed moment in time. Instead, an inclusive historiography seeks to acknowledge their dynamic history characterized by resilience and adaptation. By incorporating the generational timekeeping practices of Ifugao society and showing respect for their oral histories, we can validate the lived experiences of the community that have historically been marginalized in academic discourse.

This shift not only results in a more accurate representation of the history of the terraces but also restores agency to the Ifugao people, allowing them to define their own identity and history on their own terms. Co-author Martin, an Ifugao himself, emphasizes the importance of approaching the Ifugao and their terraces from the perspective of the Ifugao people themselves. He points out that, "We, the Ifugao, have always had a deep understanding of our past and our world. However, it was the imposition and arrogance of academic scientists and the so-called educated world that led us away from our own knowledge systems." Martin, being an Ifugao, possesses a unique ability to contextualize oral historical accounts within the cultural nuances of the Ifugao community. This expertise allows him to convey insights that may be challenging for non-Ifugao individuals to discern.

For the Ifugao, the terraces are more than just an agricultural innovation; they are a living legacy, deeply intertwined with their cultural identity, spirituality, and community life. The traditional Ifugao knowledge systems, which include intricate agricultural practices, rituals, and social structures, have been maintained and passed down through generations.

The academic scientists often approached the Ifugao and their terraces through a lens of romanticism or exoticism, at times casting them as passive relics of a bygone era rather than as active, adaptive agents of their own history. This distortion stemmed from a combination of colonialist attitudes and a lack of engagement with the community's oral histories, which offer a rich, nuanced perspective on the evolution of the terraces and the people who created and maintained them.

The imposition of these external narratives has had profound implications for the Ifugao. It has led to a disconnect between the Ifugao's self-perception and the way their history is portrayed to the world. The academic community's oversight or dismissal of Indigenous knowledge systems has not only misrepresented the terraces' history but has also undermined the Ifugao's agency in their own cultural heritage.

In recent years, there has been a growing recognition of the need to decolonize archaeology and to recenter Indigenous perspectives. This involves a conscious effort to understand the Ifugao and their terraces from the viewpoint of the Ifugao themselves. It calls for a collaborative approach that respects and incorporates Ifugao oral traditions and knowledge systems into the historical narrative. Such an approach not only corrects the historical record but also empowers the Ifugao community, allowing them to reclaim their narrative and affirm their identity on the global stage.

By valuing the Ifugao's own knowledge systems and perspectives, we can encourage a more authentic and respectful understanding of their history. This shift is crucial not only for the academic integrity of historical research but also for supporting the Ifugao in preserving their cultural heritage for future generations. The Ifugao experience thus

becomes a powerful example of how Indigenous knowledge can illuminate and enrich our understanding of the past, challenging us to rethink the way we engage with and represent the histories of Indigenous peoples.

Our work actively resists the exoticization of the Ifugao by reframing the terraces as active cultural landscapes rather than static monuments. The inclusion of local narratives in archaeological interpretation not only enriches our understanding of the past but also supports the Ifugao in safeguarding their intangible heritage. It is a reciprocal process where archaeological practice benefits from the depth of knowledge held within community stories, and the inclusion and synthesis of these stories gain visibility and validation in archaeological narratives.

This decolonized approach is evident in the way the Ifugao community has been integral to the archaeological process, from hypothesis formation to fieldwork and interpretation. Community members are not mere informants; they are collaborators with voices as authoritative as archaeologists. Their involvement in the Ifugao Archaeological Project has led to the creation of the Ifugao Community Heritage Galleries and the Indigenous Peoples Education Center, which are not only resources for cultural education but also embodiments of the community's ongoing engagement with their heritage.

The decolonization of archaeology is an ongoing and evolving practice, one that demands reflexivity, openness to change, and a commitment to justice and equity. In Ifugao, it is a conscious effort to dismantle the colonial legacy that has long influenced the interpretation of the terraces and to replace it with a collaborative, multidisciplinary approach that honors the sovereignty of the Ifugao over their cultural heritage. This method serves as a model for how archaeologists can and should move forward globally, advocating for a discipline that is inclusive, respectful, and supportive of the communities whose past we seek to understand.

This synergy between archaeological evidence and oral histories is crucial for a more inclusive and accurate portrayal of the past. It emphasizes the need to preserve these oral traditions as integral components of cultural heritage, just as vital as the physical structures themselves. Furthermore, it provides a model for other archaeological endeavors worldwide, advocating for a collaborative approach that respects and utilizes the knowledge embedded within local communities. As a result, such a confluence of perspectives not only deepens historical understanding but also empowers the descendant communities by centering their voices in the narratives of their own past [6,14].

### 1.3. Ethnographic Accounts and Community Stories

We approached ethnographic accounts in Ifugao with a strong focus on cultural sensitivity and ethical considerations. We particularly underscored the respect and the amplification of the voices and perspectives of the Ifugao community while conducting research and interpreting ethnographic accounts. We understand that the Western-centric interpretation of the terraces have long romanticized and frozen the Ifugao in time, obscuring their dynamic history of resilience and adaptation. Ethical considerations guided every step of our research and actively engaged with the Ifugao community, forging partnerships that prioritized the community's interests and autonomy.

For instance, understanding the passage of time through generational reckoning and genealogical continuity is important in comprehending Ifugao history and culture. In Ifugao society, time is traditionally measured not in years, as per the Gregorian calendar, but in generations. This method of timekeeping is central to their collective memory and is exemplified in narratives concerning the origins of significant landmarks such as the Batad Rice Terraces—one of the five clusters recognized by UNESCO—the extensive terraces in Asipulo, and the community history of Tokak Village in Namal, Asipulo.

The oral histories of the Ifugao provide a glimpse into the chronology of their landscape and culture, particularly through the narratives surrounding the Batad and Asipulo terraces. Local lore, rich in detail and passed down through generations, tells the story of two brothers from Cambulo, a village near Batad. Their discovery of the Batad hillside led

to the establishment of a swidden field, marking the beginnings of what would become a significant agricultural and cultural site. Subsequently, one brother brought his family to settle in the area, and together they constructed the terraces that have since become emblematic of Ifugao's rich tradition of rice cultivation. This story, believed to have taken place within the last six generations, illustrates a living history that is intrinsically linked to the land and the community's ancestral lineage.

In a similar vein, the Asipulo terraces, as recounted in the oral history of the Tokak community, were constructed within the past five generations. These narratives underscore a relatively recent and rapid development of terrace farming in these regions, contrasting with the often static and ancient depictions found in external historical accounts.

The story of the Tokak village exemplifies the impact of Spanish colonial expeditions on the Ifugao's settlement patterns. Village elders recount how their great grandparents, seeking to maintain autonomy and protect their way of life, were compelled to leave Amduntog—a village closer to colonial centers—and resettle in the more secluded village of Tokak. This strategic relocation allowed the Ifugao to evade direct colonial control and preserve their cultural practices, including rice terrace farming, which has been central to their identity and sustenance.

The return of their descendants to Amduntog, following the departure of the Spaniards, signifies not just a physical return to ancestral lands but also a cultural reclamation. It underscores the Ifugao's enduring connection to their territory and the resilience of their sociocultural systems amid colonial disruptions. These community stories, passed down through generations, are critical to understanding the nuanced history of the Ifugao. They provide context to the archaeological evidence, enriching our perception of the past and offering a more holistic view that honors the lived experiences and agency of the Ifugao in shaping their history.

The generational approach to history allows for a fluid and evolving understanding of the past, one that is directly connected to the memories and experiences of the community. Such oral traditions not only serve as historical records but also as affirmations of the community's enduring connection to their environment and their continuous adaptation to it. Through these stories, the people of Ifugao maintain a vibrant and ongoing relationship with their history, one that is characterized by resilience, familial bonds, and a deep respect for the terraced landscapes that have sustained them.

Dismissing these oral histories undermines the validity of ethnographic methods and shows disrespect to the Ifugao's worldview, as articulated through their community stories. These narratives align closely with the broader Philippine history, as recorded in both archival documents and contemporary scholarship. They are not merely stories; they encapsulate the lived experiences of the Ifugao.

Contrary to the notion that the Ifugao rice terraces are 2000 years old—an assumption unsupported by any archaeological evidence—recent archaeological and ethnographic data from five major sites including Old Kiyyangan Village, Hapao, Nagacadan, Batad, and Banaue suggest a much shorter history. In addition, Maher's decade-long work in the region (with at least seven sites dated) does not provide any support for the 2000-year origin of the terraces. The absence of evidence should not be mistaken for evidence of absence; yet the lack of archaeological support for the terraces' ancient origins cannot be ignored.

The primary objective of our research in Ifugao goes beyond merely dating the terraces. The terraces exemplify humanity's ability to negotiate environmental constraints and adapt to changing needs, as wet-rice cultivation is integral to Ifugao culture. These terraces represent not just a triumph over environmental challenges but also the aspirations and identities of the Ifugao. It is essential that collective consciousness moves in the direction of fostering an appreciation for the terraces not as static relics of the past but as living landscapes reflecting the ingenuity and resilience of their builders.

The convergence of archaeology with community narratives sheds light on the shared history of human societies and their environments. Archaeological evidence provides

tangible proof of past human interactions with landscapes, while community stories offer nuanced insights into these interactions. The Ifugao Archaeological Project corroborates and enriches community narratives, offering a depth of temporal understanding that enhances our appreciation of cultural practices and ecological wisdom.

In the context of climate change, this synergy becomes a valuable resource. Insights from archaeology combined with community stories offer a comprehensive view of historical climate patterns, adaptive strategies, and resilience. This knowledge is crucial for creating climate adaptation and mitigation strategies that are both scientifically sound and culturally sensitive. Understanding how communities have historically navigated environmental challenges provides invaluable lessons for today's climate action.

The integration of archaeological findings and Indigenous narratives particularly shines in the Ifugao Rice Terraces' context. It reveals a dynamic history of sustainable adaptation to climatic challenges, with Indigenous knowledge guiding effective water and soil management techniques. This Indigenous wisdom, embedded in local stories and integrated with the landscape's history, lays a robust foundation for climate-resilient agricultural practices. Moreover, this collaboration extends to natural resource management, evident in the Ifugao's *muyung* traditional forestry system [15,16].

Archaeology paired with Indigenous narratives offers insight into historical forestry practices, sustainable resource extraction methods, and community-led conservation efforts. This holistic approach recognizes the ecological wisdom encoded in Indigenous knowledge, offering vital lessons for contemporary climate adaptation.

Embracing Indigenous knowledge in climate adaptation strategies acknowledges its role as a repository of strategies finely attuned to environmental nuances. The collaborative efforts of archaeology and Indigenous knowledge bridge historical resilience with modern scientific insights, fostering a culturally attuned, sustainable, and wisdom-rooted climate adaptation paradigm.

Finally, revising the narrative to reflect the recent origins of the terraces empowers Ifugao communities, dispelling colonial myths and promoting heritage conservation programs developed and implemented by the communities themselves. Debates over the dating of the terraces risk exoticizing the Ifugao and romanticizing the past. While dating is essential, it should be secondary to understanding the terraces' cultural context.

*1.4. Nature of Archaeology: Hypothesis Testing*

The importance of two specifically dated sites in rethinking the dating of the Ifugao Rice Terraces underscores the hypothesis-testing nature of archaeology, a discipline that relies on empirical evidence to challenge, refine, or confirm existing theories. In the context of the Ifugao Rice Terraces, the traditional narrative supported by early 20th-century anthropologists posited that these terraces were over 2000 years old, a claim that has profound implications for understanding the cultural and historical landscape of the Ifugao region. The discovery and subsequent dating of two key sites have been instrumental in challenging this long-held belief, demonstrating the dynamic and evolving practice of archaeology as a field that continuously tests hypotheses against emerging data.

Archaeology, at its core, is an empirical science that constructs narratives about the past based on material evidence. Hypotheses about human behavior, cultural practices, and historical timelines are formulated based on available data, which are then tested through further excavation, analysis, and interdisciplinary research. The process is iterative: new findings can corroborate, refine, or completely overturn existing theories, leading to a more nuanced understanding of the past.

Archaeology, often misconstrued as a quest for definitive truths about our past, is fundamentally a discipline grounded in the formulation and testing of hypotheses. Rather than seeking irrefutable facts, archaeologists engage in a dynamic process of hypothesizing, gathering evidence, and revising understandings based on new findings. This approach is illustrated in the recent archaeological work on the Ifugao Rice Terraces in the Philippines,

which has led to a significant re-evaluation of their age and origins. For detail on the work of the IAP, please visit www.ifugao-archaeological-project.org.

For many years, the prevailing belief, supported by early anthropological studies, was that the Ifugao Rice Terraces were over 2000 years old. This supposition was not only an academic conjecture but also a part of the national and cultural identity of the Philippines. However, the essence of archaeological inquiry is not to cement such narratives as absolute truths but to continuously test and reevaluate them in light of new evidence and methodologies.

Acabado's research, a continuation of his graduate work [3,15–17], presented a contrasting hypothesis: the terraces were built much later, possibly as a response to Spanish colonization in the 16th century. This proposal was not an attempt to uncover a 'final truth' but rather to challenge the existing narrative based on emerging data and perspectives. The hypothesis was tested through various methods, including radiocarbon dating, archaeobotanical analysis, and ethnohistorical research, all standard tools in the archaeologist's kit for hypothesis testing.

The Ifugao Archaeological Project focused its investigations on the Old Kiyyangan Village (OKV), a site integral to Ifugao history and mythology. The scientific approach taken by the IAP was methodical and evidence based. They uncovered artifacts, such as locally produced and imported beads, a crocodile tooth, and infant burials, which provided insights into the Ifugao's interactions and adaptations over time. These findings were crucial in testing the new hypothesis about the timing and nature of terrace construction.

To test this hypothesis, the team employed a range of methods. Radiocarbon dating of charcoal samples, analysis of pollen and phytoliths, and careful stratigraphic excavation provided data points. Each method offered a piece of the temporal puzzle, contributing to a body of evidence that was then evaluated against the hypothesis.

The results of these tests were surprising. Instead of supporting the millennia-old narrative, the evidence pointed to a much more recent construction, aligning with the period of Spanish contact. This finding challenged the traditional view but was consistent with the nature of scientific inquiry, which is characterized by its openness to change when faced with new, robust evidence.

The shift in understanding about the age of the Ifugao rice terraces underscores a fundamental aspect of archaeology: it is a discipline more about asking the right questions and less about finding definitive answers. Archaeology does not deal in absolutes; rather, it embraces uncertainty and change as integral parts of understanding the past. Each new piece of evidence can redefine existing narratives, demonstrating that our understanding of history is fluid and subject to revision.

This perspective is crucial to considering the role of archaeology in society. By not positioning itself as a seeker of absolute truth, archaeology allows for multiple narratives and interpretations to coexist. This approach is particularly important in areas like the Ifugao province, where cultural identity and historical narratives are deeply intertwined. The re-evaluation of the terraces' age does not diminish their cultural and historical significance; instead, it enriches our understanding of the Ifugao people's resilience, adaptability, and agency.

Moreover, the case of the Ifugao Rice Terraces highlights the importance of integrating scientific inquiry with local knowledge and oral histories. The Ifugao community's narratives and memories played a crucial role in shaping the hypothesis and guiding the archaeological investigation. This integration underscores the importance of a holistic approach in archaeology, one that respects and incorporates various sources of knowledge.

The evolving understanding of the Ifugao rice terraces' history exemplifies the nature of archaeology as a discipline focused on testing hypotheses rather than seeking immutable truths. This case study demonstrates how archaeology is a dynamic and iterative process, one that adapts and advances as new evidence emerges. It reminds us that our grasp of the past is always provisional, and that openness to re-evaluation and revision is essential in

the pursuit of knowledge. In this way, archaeology not only helps us to understand our history but also teaches us about the complexity and fluidity of the human experience.

The Ifugao case exemplifies the iterative nature of archaeological research. Hypotheses are constructed based on the best available information, then tested against empirical data. If the data align, the hypothesis is strengthened; if not, it must be revised or discarded. This process is not unique to archaeology but is the hallmark of the scientific approach across disciplines.

Through this example, we see that archaeology, like all science, is a dynamic process. It is a discipline that evolves with each new discovery, where hypotheses are continuously refined to enhance our understanding of the human past. Far from seeking immutable truths, archaeology embraces the complexity and variability of human history, offering insights that are as provisional as they are profound.

## 2. Materials and Methods

The recent archaeological work in Ifugao, Philippines, continues and expands upon Stephen Acabado's prior research [3,10,15–18], suggesting the iconic Ifugao rice terraces were constructed in response to Spanish colonization. This challenges the older belief that the terraces are millennia old and reshapes the understanding of the region's history and the Indigenous Ifugao people. The AMS determinations and spatial analyses are also supported by archaeobotanical datasets. There is a total absence of any evidence of wet-rice cultivation in the region that predates the 1600s.

The Ifugao Archaeological Project's (IAP) focus on the Old Kiyyangan Village (OKV) has been pivotal in uncovering the area's past. Fray Molano, in a Spanish document from 1801, described OKV as a large settlement—a contrast to its portrayal as isolated and unchanging. Robert Maher's [19] initial research at OKV provided important chronological data, indicating the site was much older than previously understood, with findings suggesting Ifugao ancestors settled there.

The selection of OKV for archaeological exploration was driven by the community's desire to investigate their origins. Despite its transformation into paddy fields, historical records from the American colonial period mention the Village of Otbobon, another name for OKV, suggesting a relocation by 1869 [20]. The modern town of Kiangan's proximity to the original OKV site highlights shifts in settlement patterns.

Robert Maher's [19] initial investigations into OKV, now a rice field believed to have been transformed into a paddy field just before World War II, provided important chronological data. Maher's excavation yielded two thermoluminescence (TL) dates, ranging between 1130 CE and 1230 CE, indicating a much earlier occupation of the site than previously understood (Table 1). This finding is crucial, as OKV is deeply ingrained in the cultural and mythical history of the Ifugao. According to local myths, Old Kiyyangan was the village where the first Ifugao ancestors settled and where they received divine knowledge of wet-rice cultivation. These narratives also suggest that prior to the development of the terraces, taro was the primary crop cultivated by the Ifugao. Maher's research in the region set out to provide archaeological evidence for the origins of the Ifugao and their rice terraces [21–24], which provided the initial radiocarbon dates (Table 2).

**Table 1.** Maher's TL dates in the OKV Site and adjacent Bintacan Cave.

| Site | Level Info | TL Dates | Reported by |
|---|---|---|---|
| Bintacan Cave | Level F | 1620 BP Alpha 476 | [24] |
| Bintacan Cave | Level E | 1420 BP (±20%) Alpha 480 | [24] |
| Bintacan Cave | Level C | 760 BP (±20%) Alpha 479 | [24] |
| Old Kiyyangan Village | No data presented | 820 BP Alpha 566 | [20] |
| Old Kiyyangan Village | No data presented | 720 BP Alpha 671 | [20] |

**Table 2.** Radiocarbon determinations from Banaue Ifugao, obtained by Maher [21,22] and Harold Conklin [25]. While some of these radiocarbon determinations show older (>500 years), these samples are not contextualized and may not be associated with the archaeological event in question. For more discussion, refer to Acabado and colleagues' discussions [3,5].

| Site/Locality | Depth (cm) | Lab # | Material | ¹⁴C BP | Cal. CE (2 σ–95%) (Recalibrated Using IntCal13) | ¹⁴C Sample Context | Reported by |
|---|---|---|---|---|---|---|---|
| If1-Nabyun | 91 | GX0668 | *M. sinensis* | 205 ± 100 | 1493-postbomb | Pond-field | [21] |
| If2-Nabyun | 91 | GX1900 | No data | 325 ± 110 | 1408-postbomb | House Platform | [21] |
| If2-Nabyun | 90 | GX1901 | No data | 695 ± 100 | 1052–1435 | Midden | [21] |
| If2-Nabyun | 90 | BX2184 | No data | 735 ± 105 | 1043–1413 | Midden | [21] |
| If3-Banaue | 4.4 | GX2183 | No data | 2950 ± 250 | 1867-540BCE | House Platform | [21] |
| Gawwa, Poitan | 5 | GX3138 | No data | 530 ± 140 | 1192–1792 | Underground chamber | [21] |
| Gawwa, Poitan | 5 | GaK5238 | No data | 530 ± 100 | 1273–1631 | Underground chamber | [21] |
| Lugu | No data | UGA2515 | No data | 395 ± 60 | 1430–1639 | Terrace embankment post | [25] |
| If20-Banghallan | 50 | GaK6442 | No data | 890 ± 310 | 434–1647 | Village edge | [22] |
| If20-Banghallan | 60 | UGA1541 | No data | 1340 ± 375 | 176BCE-1388CE | Village edge | [21] |

Between 2012 and 2016, the IAP executed twenty excavation units at OKV. The findings from these excavations have been revealing, indicating that the arrival of the Spanish in the region was a catalyst for significant changes in the village. Most notably, the shift to wet-rice cultivation seems to have occurred concurrently with the Spanish influence, supported by AMS dates. The excavations in OKV unearthed a variety of artifacts, including locally produced and imported beads, a crocodile tooth, imported metal adornments, and infant burials [26–31]. These discoveries are indicative of the active role the Ifugao community played in broader pre-colonial and colonial interactions within the Philippines, challenging the notion of their cultural and economic isolation. The initial phase yielded seven radiocarbon dates from the terraced fields of Banaue (Table 3), and subsequent excavations at the Old Kiyyangan Village (OKV) added nineteen more dates (Table 4). These were complemented by extensive analyses, including sherd residue [29], wood charcoal, and microfossil studies, to determine the advent of wet-rice agriculture in the region.

**Table 3.** AMS dates from Bocos rice terraces, Banaue, Ifugao [3].

| Lab. No. | Unit | DBS | Layer | CRA | ¹³C | Cal AD (BCal) | Post-AD 1585 Probability * |
|---|---|---|---|---|---|---|---|
| AA78973 | Mamag | 0.855 | II | 119 ± 38 | 25.2 | 1687–1862 | 74.6% |
| AA78974 | Mamag | 1.3 m | III | 485 ± 39 | −27.5 | 1325–1460 | 74.6% |
| AA78971 | Rasa | 0.35 m | II | 313 ± 38 | −24.4 | 1620–1800 | 98.5% |
| AA78972 | Rasa | 0.52 m | III | 164 ± 38 | −26.0 | 1527–1757 | 98.5% |
| AA78969 | Linagbu | 0.55 m | II | 180 ± 38 | −26.5 | 1736–1867 | 99.9% |
| AA78970 | Linagbu | 0.75 m | III | 131 ± 38 | −29.3 | 1663–1753 | 99.9% |
| AA78975 | Achao | 0.075 m | II | 193 ± 38 | −25.0 | 1646–1809 | N/A |

* Probability analyses (Bayesian modeling) of pre-Spanish or post-Spanish construction of the Bocos rice terraces walls in Banaue, Ifugao [3].

Excavations at OKV over several field seasons resulted in the opening of 21 trenches and 10 shovel test probes, uncovering over 20,000 artifacts, predominantly earthenware ceramics, as well as faunal, archaeobotanical samples, and human remains, mostly infant burials. Radiocarbon dating at OKV employed diverse materials—bulk soil, wood charcoal,

bone collagens, and sherd residue—all showing a consistent stratigraphy. The alignment of bulk soil dates within situ radiocarbon dates, despite potential contamination risks, suggests rapid deposition within a short timeframe.

**Table 4.** Radiocarbon determinations recovered from Ifugao between 2012 and 2016 [5].

| Depth (cm) | Lab Number | Location | Material/Trench | $^{14}$C BP | Cal. BP (2 σ) | Cal. CE (2 σ) | Context |
|---|---|---|---|---|---|---|---|
| 30–40 | Beta-356307 | OKV | organic sediment/8 | 190 ± 30 | 260–200 | 1640-post-1950 | Rice field |
| 50–60 | UCIAMS-183276 | OKV | Wood charcoal (*P. insularis*)/14 | 415 ± 15 | 510–469 | 1440–1480 | Rice field |
| 55–73 | Beta-394185 | OKV | bone collagen/8 | 410 ± 30 | 530–470 | 1405–1445 | Mortuary |
| 60–70 | UCIAMS-183272 | OKV | Wood charcoal (*P. insularis*)/14 | 345 ± 15 | 477–317 | 1470–1633 | Rice field |
| 65–70 | Beta-356306 | OKV | organic sediment/8 | 620 ± 30 | 680–620 | 1280–1390 | Rice field |
| 80–90 | UCIAMS-183273 | OKV | Wood charcoal (*P. insularis*)/14 | 570 ± 15 | 634–537 | 1315–1415 | Rice field |
| 80–90 | Beta-394182 | OKV | bone collagen/8 | 600 ± 30 | 730–670 | 1265–1380 | Mortuary |
| 90–100 | CIAMS-183274 | OKV | Wood charcoal (*P. insularis*)/14 | 665 ± 15 | 669–564 | 1280–1385 | Rice field |
| 90–100 | Beta-421036 | OKV | charcoal/14 (*P. insularis*) | 660 ± 30 | 690–630 | 1280–1390 | Rice field |
| 90–100 | Beta-421037 | OKV | potsherd residue/14 | 590 ± 30 | 610–550 | 1300–1415 | Rice field |
| 90–100 | D-AMS 003446 | OKV | Organic sediment/9 | 861 ± 25 | 899–700 | 1052–1250 | Rice field |
| 100–110 | D-AMS 003447 | OKV | Organic sediment/10 (fill) | 1252 ±37 | 1279–1075 | 672–876 | Rice field |
| 100–110 | D-AMS 003448 | OKV | Organic sediment/10 (dark midden soil) | 292 ±27 | 456–291 | 1495–1660 | Rice field |
| 100–110 | Beta-356305 | OKV | organic sediment/8 | 720 ± 30 | 810–750 | 1220–1280 | Rice field |
| 110–120 | D-AMS 003445 | OKV | Organic sediment/9 | 672 ± 28 | 676–561 | 1274–1390 | Rice field |
| 110–120 | Beta-32953 | OKV | organic sediment/3 | 780 ± 30 | 741–669 | 1160–1260 | Rice field |
| 120–144 | Beta-394184 | OKV | bone collagen/9 | 800 ± 30 | 767–675 | 1045–1220 | Mortuary |
| 130–140 | Beta-329552 | OKV | organic sediment/3 | 770 ± 30 | 734–668 | 1050–1240 | Rice field |
| 150–160 | Beta-329551 | OKV | organic sediment/3 | 1000 ± 30 | 967–799 | 900–1020 | Rice field |

Paleoethnobotanical analysis indicated a potential presence of wet-rice cultivation dating back to 675 years before present (BP), but low counts of rice phytoliths suggested possible displacement within the soil column due to water seepage [32,33], casting doubt on earlier cultivation theories. Residue analysis from pottery did not indicate rice production, but instead showed evidence of taro consumption, further supporting the argument for the later emergence of wet-rice agriculture in Ifugao.

Analyses of bulk soil, charred residues, and botanical data from three excavation trenches suggest that the predominant crop in the OKV prior to 1650 CE was not wet rice, as previously believed, but taro (*Colocasia esculenta*). The absence of wet-rice cultivation before the mid-17th century was corroborated by pollen, phytolith, and starch analyses performed on sherd residues from these trenches, further dismantling the theory of an ancient origin for rice agriculture in the region. Details on the microbotanical analyses from soils obtained in the OKV site are provided by Horrocks et al. [34] report and Acabado's preliminary hypothesis [35,36].

Additionally, the stratigraphy of the site revealed a significant increase in wood charcoal within Layer 2, dated to around the 1600s (Figure 3). This finding could indicate a heightened demand for wood or point to deforestation, reflecting a change in landscape management or environmental conditions. Crucially, no rice or rice-associated weeds were detected in any soil or charred residue samples predating 1650 CE, supporting the hypothesis of a post-contact introduction of wet-rice agriculture.

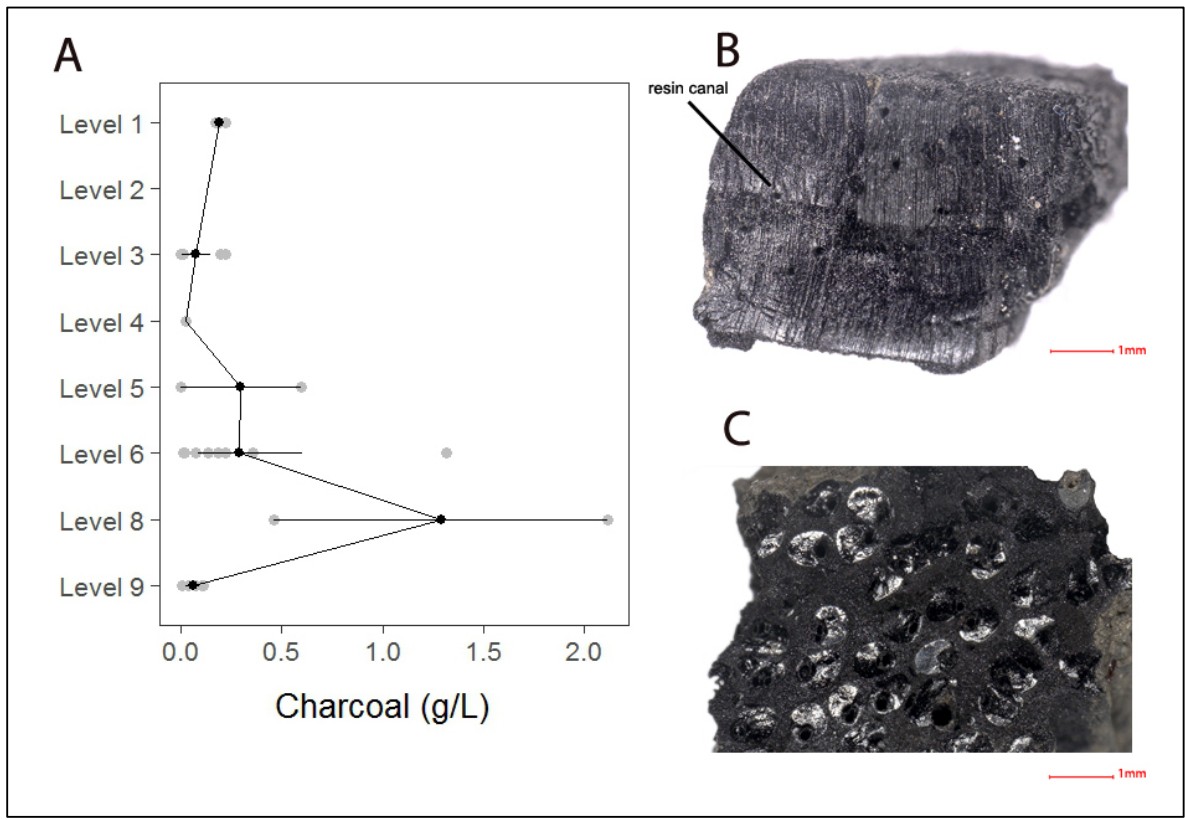

**Figure 3.** (**A**) The densities of wood charcoal (grams per liter) recovered in deposits from Trench 14, where the level (i.e., depth in 10 cm increments) is represented on the vertical axis, and the density is represented on the horizontal axis. The gray dots illustrate the values in individual samples, and the black dots are the bootstrapped mean densities of each level with 95% confidence intervals (the black line within a level). The line between levels connects the means of each level. (**B**,**C**) are the transverse (cross) sections of identified wood charcoal, where (**B**) is identified as Benguet pine with a resin canal highlighted from Trench 14, level 8, and (**C**) is a specimen from the palm family from Trench 16, level 2 [5]. Figure provided by A. Farahani.

The environmental transformation indicated by the archaeological record was also marked by a notable influx of imported goods, a surge in the remnants of animals used in rituals, and the adoption of wet-rice farming practices. These shifts signal a broader change in the Ifugao way of life, aligning with the period of increased external contact and trade [36].

This multi-evidentiary approach has been crucial in constructing a robust chronological model for the inception and expansion of the Ifugao terracing systems. Ethnographic comparisons highlight that wet-rice cultivation necessitates complex social structures. These findings support the hypothesis that the Ifugao terraces, as seen today, were constructed within a 200- to 300-year period, a rapid development indicating advanced sociopolitical organization suitable for intensive wet-rice farming.

## 3. Results

The lack of evidence for the 2000-year-old origin of the terraces across five major Ifugao sites has led to the discrediting of this long-held belief, with recent archaeological and ethnographic data suggesting a much shorter history. The significance of local time reckoning in Ifugao, which relies on generational memory rather than calendar years, has been crucial in contextualizing this narrative shift. Community stories from the Batad Rice Terraces, Asipulo terrace system, and Tokak Village provide a vivid account of the terraces' origins tied closely to the Ifugao's response to Spanish colonialism, rather than a distant, pre-colonial past.

Archaeobotanical and pottery residue data have revealed the continuation of taro as a staple carbohydrate, even after the Spanish arrival in the adjacent lowlands of the Magat Valley. The maintenance of traditional burial practices and dietary habits through faunal isotopic signatures indicates that the Ifugao retained their way of life despite the external pressures of colonialism.

These datasets collectively suggest that the Ifugao responded rapidly and adaptively to Spanish incursion, upholding their group identity and cohesion through ritual and household practices. The AMS determinations and spatial analyses of the terraces, alongside archaeobotanical datasets, point to an absence of pre-1600s wet-rice cultivation, challenging the narrative of an ancient terracing tradition.

The Ifugao terraces stand not only as a testament to humanity's ability to adapt to environmental constraints but also as a reflection of the Ifugao's identity and resistance against colonial forces. The implications of these findings are not limited to local history but contribute to broader discussions on agricultural systems' archaeology, requiring a comprehensive approach that encompasses local knowledge and generational narratives.

Spanish colonial documents recorded active irrigation systems in only two regions of the Philippines during the early years of conquest: Bicol and the Magat Valley. No other archaeological or historical investigations have documented subsistence patterns before and after the Spanish conquest. Therefore, while dating the terraces is crucial, it is secondary to the broader aim of understanding the terraces' role in Ifugao culture and heritage.

In the neighboring region of Bontoc, Bodner's [37] work has provided compelling evidence for the later inception of wet-rice cultivation, postdating 1600 CE. This assertion is grounded in the notable absence of archaeobotanical data indicative of wet rice prior to the 1600s. Supporting this perspective, an accumulation of data from various scholarly pursuits—including Maher's investigations, Bodner's own dissertation, Conklin's landscape research, and the ongoing Ifugao Archaeological Project—converge to suggest a significant shift in agricultural practices to wet-rice cultivation occurring relatively recently.

Further strengthening this argument, a genomic study of rice varieties in the Cordillera, including the Ifugao's revered tinawon rice, has shed light on the crop's lineage. According to Alam et al. [38], the highland rice varieties can trace their ancestry to Indonesian strains, suggesting a southern origin and diffusion into the highlands. This genomic lineage contradicts the theory of an ancient, Indigenous development of wet-rice cultivation in the Ifugao region, instead pointing to a more recent introduction and adaptation of these rice varieties. Together, these interdisciplinary research efforts paint a picture of dynamic agricultural evolution in the Cordillera, characterized by later adoption and continuous innovation.

This body of work underscores the need for data-driven discussions about the dating of the Ifugao terracing tradition, advocating for a narrative that recognizes the terraces as a dynamic cultural landscape shaped by historical contingencies and the Ifugao's resilient spirit.

## 4. Discussion: Archaeology and Community

Comprehensive archaeological investigations spearheaded since 2007 have significantly reshaped the narrative of the Ifugao Rice Terraces. This intensive research, however, is more than a mere quest for chronological accuracy; it is an exploration into the lived experiences of the Ifugao people, as evidenced by both material culture and enduring oral

traditions. The integration of archaeological findings with local narratives has revealed a history marked not by stasis and isolation but by dynamism, resilience, and adaptation.

The ethnographic accounts and community stories of the Ifugao stand as testaments to the society's active participation in their history. These narratives, passed down through generations, serve not only as historical records but also as expressions of the Ifugao's identity and response to external pressures, particularly Spanish colonization. The recollections of the origins of landmarks such as the Batad Rice Terraces and the Asipulo terrace system underscore a temporal understanding deeply ingrained in Ifugao culture. Moreover, the tales of strategic relocations during colonial times, such as those from Amduntog to the more remote Tokak, illustrate the Ifugao's agency in preserving their way of life.

The archaeological record, bolstered by radiocarbon dating and analyses of pottery residue and paleoethnobotanical samples, has supported the contention that the terraces are of more recent origin than previously thought. These data have challenged the long-held belief that the terraces are over two millennia old, a belief unsupported by evidence from significant sites within the region. The findings suggest a rapid socioeconomic transformation in Ifugao society, coinciding with the emergence of wet-rice agriculture and the introduction of new goods and practices, indicating an era of significant change catalyzed by, but not succumbing to, colonial influence.

This cultural chronology is pivotal in understanding the terraces not as static monuments but as vibrant landscapes that encapsulate humanity's enduring adaptability. The terraces are symbols of the Ifugao's triumph over environmental and political challenges, embodying the aspirations and ingenuity of their creators. Recognizing the terraces as part of a living culture is essential for fostering respect for Ifugao heritage and for ensuring the terraces' preservation for future generations.

The role of Indigenous knowledge, particularly the *muyung* traditional forestry system, further exemplifies the Ifugao's sustainable management of natural resources [15,16]. The confluence of archaeological evidence with this Indigenous wisdom offers insights into historical practices that can inform contemporary climate adaptation strategies. Embracing this knowledge is crucial for crafting solutions that are not only effective but also respectful of cultural traditions.

Moreover, by integrating community narratives with archaeological research, a more nuanced picture of the Ifugao's past emerges. These stories provide context and color to the archaeological canvas, enriching our understanding of the terraces and the people who built and maintained them. The Ifugao Archaeological Project has shown that the terraces' recent origins do not diminish their value; rather, they highlight the Ifugao's remarkable adaptability and the terraces' ongoing significance as cultural and ecological marvels.

As global discussions on climate action and cultural conservation continue, the insights gleaned from the Ifugao terraces serve as a valuable resource. The Ifugao's historical experiences, coupled with archaeological data, provide a model for understanding how societies can navigate environmental challenges while maintaining their cultural integrity.

The narrative of the Ifugao rice terraces is being rewritten. It is a narrative that respects the intersection of archaeology and oral history, acknowledges the complex interplay between the Ifugao and their environment, and recognizes the terraces as a symbol of cultural resilience. This narrative shift has profound implications for heritage conservation, climate action, and our understanding of humanity's relationship with the land. As we move forward, it is essential to continue valuing and incorporating the voices of the Ifugao community, ensuring that their history is told with accuracy and respect, and that their wisdom is heeded in the face of contemporary challenges.

## 5. Conclusions: Changing the Narrative, Empowering Descendant Communities

The Ifugao Archaeological Project demonstrates a successful integration of community perspectives with archaeological data, a practice that has been pivotal in reshaping the narrative of the Ifugao Rice Terraces. By engaging local narratives and participation, this project has moved beyond the confines of traditional, colonial methodologies that often

perpetuate a romanticized and static view of Indigenous communities. Instead, it has revealed the terraces as living emblems of the Ifugao's resistance and adaptability through centuries of sociocultural transformations. An example of this knowledge co-production is highlighted by this video link: The Old Kiyyangan Village Story.

This project has catalyzed a re-evaluation of the terraces' history, aligning archaeological findings with the oral histories of the Ifugao. This has disrupted the academic and popular portrayal of the terraces as ancient and untouched by historical events, instead highlighting the Ifugao's active role in their evolution. The Ifugao's own generational timekeeping and narratives offer an invaluable insight into the cultural significance of the terraces, detailing a history of dynamism, community cohesion, and resilience against colonial pressures [39,40].

Furthermore, the IAP's collaborative approach has empowered the Ifugao community, allowing them to reclaim their narrative and cultural heritage. The development of the Ifugao Community Heritage Galleries and the Indigenous Peoples Education Center (Figure 4) exemplifies the tangible benefits of this inclusive practice. These centers not only serve as repositories of Ifugao material culture but also as educational resources that enable the development of local history curricula, thereby preserving Indigenous knowledge for future generations.

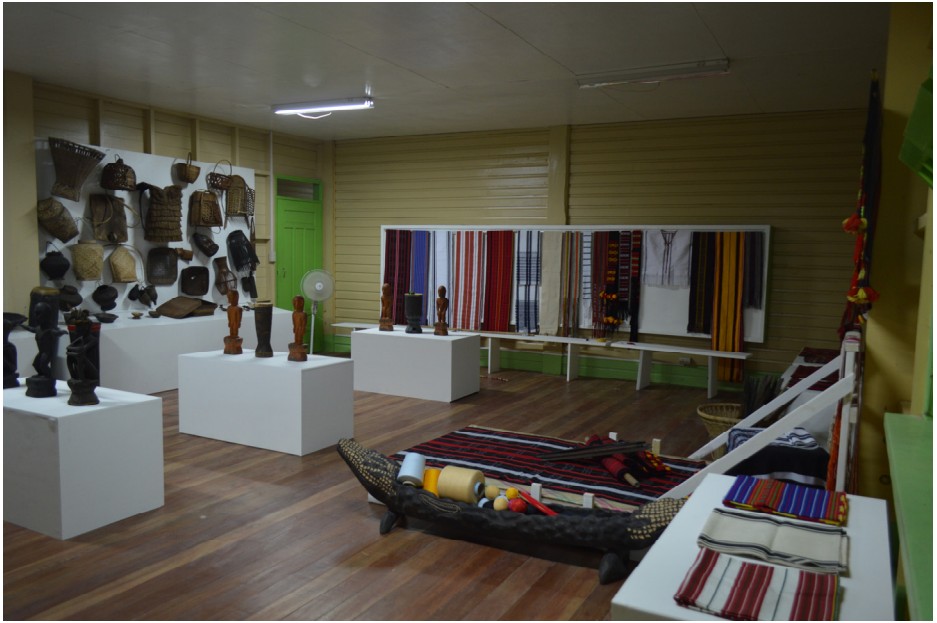

**Figure 4.** The weaving exhibit at the Ifugao Community Heritage Galleries, which now serve as the Ifugao Indigenous Peoples Education Center.

Such practices exemplify the decolonization of archaeology, defined as the process of re-evaluating and transforming practices to remove colonial biases and structures. Decolonization in archaeology prioritizes Indigenous perspectives and knowledge systems, seeking to correct historical power imbalances in the interpretation and stewardship of cultural heritage. This approach emphasizes the need for descendant communities to be actively involved in research processes, ensuring their oral traditions and sovereignty over their cultural narratives and material past are respected and integrated into the historical narrative.

The Ifugao experience showcases how the integration of community voices with archaeological research can lead to a more accurate and respectful understanding of the past. It demonstrates that the decolonization of archaeology is not only possible but necessary for the discipline to evolve in a way that is inclusive, equitable, and just. The IAP exemplifies the paradigm of community archaeology, where local narratives and participation fundamentally reshape the interpretation and conservation of cultural heritage. This inclu-

sive approach has illuminated the Ifugao rice terraces not merely as UNESCO-recognized aesthetic monuments but as living emblems of resistance against colonial imperialism—a legacy enduring through centuries of sociocultural transformations.

Community archaeology in Ifugao is not an isolated phenomenon. Around the world, similar movements have been gaining momentum, recognizing the value of incorporating local voices into archaeological narratives. In Australia, Indigenous archaeology has become a vital practice, with Indigenous Australian communities working alongside archaeologists to protect and interpret their ancestral lands [41–43]. This collaboration has enabled a deeper understanding of the Indigenous Australian history and culture, which spans tens of thousands of years.

Moreover, in northern Africa, community archaeology projects have engaged with local communities to explore pre-colonial history, often overshadowed by the focus on ancient monuments like the pyramids of Egypt e.g., [44,45]. These projects not only uncover rich local histories but also empower communities by involving them in the preservation and interpretation of their heritage.

The significance of community participation in archaeology is multifaceted. It democratizes the process of historical inquiry, acknowledges the enduring presence of Indigenous cultures, and ensures that heritage conservation is not just an academic exercise but a communal endeavor. By combining traditional knowledge with archaeological findings, communities gain a voice in telling their own stories—a process that validates their history and identity.

The IAP's community-based approach also has broader implications for global heritage conservation. By demonstrating the tangible benefits of inclusive practices, it sets a precedent for similar initiatives worldwide. It underscores the importance of local engagement in preserving not only the physical remnants of the past but also the intangible aspects of culture that give meaning to these relics.

In the Ifugao case, the terraces are more than agricultural feats; they are a testament to a people's ingenuity and resilience. The community's involvement in the IAP has revitalized interest among the younger generation, inspiring them to delve into their history and to value the disciplines of anthropology and archaeology. This renewed interest has the potential to create a cadre of young Ifugao who are not only aware of their cultural heritage but are also equipped to preserve and promote it.

The integration of community voices with archaeological expertise in Ifugao provides a model for how heritage sites around the world can be managed and interpreted. It shows how the past can be a source of pride and identity for contemporary communities and how archaeology can serve as a bridge between the past and the present. This collaborative model can inspire similar efforts globally, where local communities become active participants in the conservation and interpretation of their cultural heritage.

*Inclusive National Identity*

Our approach to the archaeological practice in Ifugao offers a compelling example of decolonizing methodology that aligns with Indigenous movements in the Philippines. Our work challenges the traditional colonial narratives that have often defined the history and significance of sites like the Ifugao Rice Terraces, providing a framework for understanding these landscapes that is rooted in the community's own history and knowledge systems.

We emphasize the importance of community engagement in archaeological research. This involves integrating Indigenous knowledge and oral histories with archaeological methods to construct a more holistic understanding of the past. By doing so, we not only question the colonial underpinnings of historical narratives but also elevate the role of the Ifugao people as custodians of their own history.

The decolonial perspective offered by our work engages with earlier discourses on Indigenous peoples' movements in the Philippines by validating their narratives and recognizing their agency. Instead of treating Indigenous communities as mere subjects of

research, our approach involves them as active partners, thereby disrupting the power dynamics that have traditionally characterized archaeological practice.

Furthermore, our work does not shy away from the theoretical or political implications of decolonization. While it is deeply embedded in the practical aspects of archaeological work, such as site excavation and material analysis, it also confronts the theoretical frameworks that have historically marginalized Indigenous perspectives. By advocating for a participatory approach to archaeology, we implicitly critique the political structures that have perpetuated colonial attitudes in the discipline.

In terms of engaging with broader movements for Indigenous rights in the Philippines, our decolonial approach reinforces the political aims of these movements. It recognizes the sovereignty of Indigenous peoples over their cultural heritage and supports their struggle for self-determination. The emphasis on community collaboration ensures that the fruits of archaeological research benefit the Indigenous communities themselves, whether through education, tourism, or reinforcing land claims.

Our decolonial methodology in archaeological practice is not merely an academic exercise. It is a form of activism that supports the broader goals of Indigenous peoples' movements in the Philippines. By engaging with the community and placing their knowledge at the forefront, our work exemplifies how decolonization can be both a practical and a political act, addressing the call for a more inclusive and just representation of the past.

The Ifugao case has shown that community archaeology, though not perfect and not an answer to all problems, promises to minimize conflict between heritage stakeholders. The practice of community archaeology also intensifies conversations between archaeologists and descendant communities. None of this is to suggest that community archaeology solves all of the complicated problems and compromises of archaeology and of interactions with communities with their own local, regional, and national entities. The success of any heritage management program rests on the engagement of many segments of the community.

**Author Contributions:** Conceptualization, S.B.A. and M.M.M.; methodology, S.B.A.; formal analysis, S.B.A.; investigation, S.B.A. and M.M.M.; data curation, S.B.A.; writing—original draft preparation, S.B.A.; writing—review and editing, S.B.A. and M.M.M.; visualization, S.B.A.; supervision, S.B.A.; project administration, S.B.A.; funding acquisition, S.B.A. All authors have read and agreed to the published version of the manuscript.

**Funding:** Funding for this project was generously provided by the NSF-REU Site: Ifugao Archaeological Project (Award #1460665), the National Geographic Society (NGS-9069), the Hellman Fellowship, UCLA's COR and FCDA Grants, the Institute for Field Research, and the National Museum (Philippines) Grant-in-Aid of Research.

**Data Availability Statement:** There are no new data created in this article. However, landscape (GIS) and Bayesian modeling data as presented in this manuscript can be provided by email the corresponding author.

**Acknowledgments:** This paper is the culmination of a decade of dedicated research in Ifugao, with a significant focus on the Old Kiyyangan Village, Kiangan, Ifugao, in the latter half of that period. The members of the Ifugao Archaeological Project (IAP) from 2012 to 2016 were instrumental in the acquisition and organization of the data presented herein. Collaborative efforts with the Save the Ifugao Terraces Movement, Archaeological Studies Program at the University of the Philippines, and the National Museum of the Philippines were pivotal in facilitating our fieldwork endeavors. Our gratitude extends to the Butic-Baguilat family for their generosity in allowing us access to the OKV site, despite the disruptions to their agricultural activities. Our acknowledgments would be incomplete without mentioning the insightful comments and suggestions provided by Iman Nagy, which significantly enriched the content of this manuscript. We are also indebted to three anonymous reviewers who provided constructive comments. We alone are responsible for the final product and take responsibility for any errors of fact or interpretation.

**Conflicts of Interest:** The authors declare no conflicts of interest.

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
