# Peer review of "Older Is Not Necessarily Better: Decolonizing Ifugao History through the Archaeology of the Rice Terraces"

_land, doi:10.3390/land13020237_

Round 1

Reviewer 1 Report

Comments and Suggestions for Authors

This paper challenges the traditional perspective, especially the chronology of the Ifugao rice terrace. The authors corroborated the statement based on environmental archeology and ethnographic methods. This paper fully supports their statement and has conducted a lot of researches. There are few suggestions stated as below.

Some lab works are crucial for the chronological sequence of the rice terrace, for instance, the authors stated that based on pollen, phytolith, and starch analysis from OKV, it revealed that the predominant crop in OKV before 1650CE was Taro. I suggest the authors either supply a supplementary document or add a few paragraphs to present the lab process and detailed result about pollen, phytolith, and starch analysis.

The resolution of the first figure needs to be improved.

Author Response

Reviewer 1

This paper challenges the traditional perspective, especially the chronology of the Ifugao rice terrace. The authors corroborated the statement based on environmental archeology and ethnographic methods. This paper fully supports their statement and has conducted a lot of researches. There are few suggestions stated as below.

Some lab works are crucial for the chronological sequence of the rice terrace, for instance, the authors stated that based on pollen, phytolith, and starch analysis from OKV, it revealed that the predominant crop in OKV before 1650CE was Taro. I suggest the authors either supply a supplementary document or add a few paragraphs to present the lab process and detailed result about pollen, phytolith, and starch analysis.

  • Details on the microbotanical analyses from soils obtained in the OKV site are provided by Horrocks et al. [38] report and Acabado’s preliminary hypothesis [20].

The resolution of the first figure needs to be improved.

  • Addressed, added high resolution images

Reviewer 2 Report

Comments and Suggestions for Authors

Thank you for a very interesting read. The use and integration of archaeological and local knowledge is an important and highly relevant topic, and the message the authors present is an important one.

On a general note, the paper might benefit from a more clearly defined focus/aim and some restructuring of parts of the text.

Parts of the paper are well referenced, others not.

Abstract

Concise and to the point.

Introduction

Relevant information that provides a good background and describes the issue under investigation. However, the introduction also contains a significant amount of information that reads more as concluding remarks and statements. The main point and message the authors want to get across seems to be presented in the introduction, in Decolonizing Archaeology in particular. Some parts of the introduction might serve better placed later in the paper.

The abstract starts off with describing what the study examines. The introduction does not do the same, it does not clearly state the aim and/or research questions of the paper. It would be beneficial to more clearly describe the aim or research questions. A clearly stated aim or research question might also aid in providing structure.

Regarding Figure 1.

The figure should be improved. To improve readability, it would be beneficial to separate the map and show this as a separate figure. Also, it would be beneficial to improve the resolution as the legend is all but illegible.

The photos are beneficial for readers not familiar with the Ifugao Rice Terraces. However, the resolution needs to be improved significantly. For instance, the legend on one photo is illegible. A figure/photo with a closeup of the terraces would also be of interest for readers not familiar with this heritage.

Photographer?

Ethnographic accounts and community stories

The information presented in this chapter is interesting. There is however not much information on method and methodological approaches on for instance documenting and utilizing indigenous knowledge in this context (Ifugao). Methodological and ethical consideration

The paper mentions that one of the authors is an Ifugao but does not clarify why this is important to specifically mention, e.g. as a holder of knowledge, as having a perspective that allows for a better understanding or so on.

Nature of archaeology: hypothesis testing

A link to the Ifugao Archaeological Project website might be useful for those interested in finding more information about the archaeological research and the sites.

Archaeological data

OK presentation. Well enough referenced.

Regarding Figure 2
The caption, yellow highlights of the word axis

Archaeology and community

No explanation or reference for, for instance the muyung traditional forestry system

Parts of the text read as concluding remarks and statements. A summary/conclusion?

Changing the narrative, empowering descendant communities

Line 688: he? Should it be we instead? If not, who is he?
Same for line 694 (his).

This chapter runs through several important topics quite briefly, several by using examples not Ifugao. The examples are relevant, but the relevance in terms of Ifugao and the Ifugao Rice Terraces could perhaps be an even stronger focus in this part. The various topics/sub-chapters are brief, it might be beneficial to chose fewer topics and focus more on one or two questions.

In the final paragraph, it is briefly mentioned that community archaeology is not perfect and does not answer all problems. This is an important point, but at no point prior has there been any mention or discussion of the associated challenges or weaknesses.

Regarding figure 3.

The resolution of especially one of the pictures could be better, but it is not quite as bad as figure 1.

Concerning the bottom picture of grade five students where some faces are seen clearly enough to be identifiable. Have there been any consideration concerning ethical issues regarding this? Is there parental permission to publish this picture of minors? If so, this should be made clear in the caption. However, removing the picture of the children is an option that should definitely be considered.

Photographer?

Author Response

Reviewer 2

Thank you for a very interesting read. The use and integration of archaeological and local knowledge is an important and highly relevant topic, and the message the authors present is an important one.

On a general note, the paper might benefit from a more clearly defined focus/aim and some restructuring of parts of the text.

Parts of the paper are well referenced, others not.

Abstract

Concise and to the point.

Introduction

Relevant information that provides a good background and describes the issue under investigation. However, the introduction also contains a significant amount of information that reads more as concluding remarks and statements. The main point and message the authors want to get across seems to be presented in the introduction, in Decolonizing Archaeology in particular. Some parts of the introduction might serve better placed later in the paper.

The abstract starts off with describing what the study examines. The introduction does not do the same, it does not clearly state the aim and/or research questions of the paper. It would be beneficial to more clearly describe the aim or research questions. A clearly stated aim or research question might also aid in providing structure.

  • This paper aims to demystify, and in the process, decolonize, archaeological and historical narratives developed in the 1900s by pioneer anthropologists who worked in the Cordillera region of the Philippines.
  • However, our approach challenges this long-standing narrative by actively involving the Ifugao community in the research process. By doing so, we uncover a more nuanced and historically grounded understanding of the Ifugao Rice Terraces and the people who have shaped their history over generations.

Regarding Figure 1.

The figure should be improved. To improve readability, it would be beneficial to separate the map and show this as a separate figure. Also, it would be beneficial to improve the resolution as the legend is all but illegible.

The photos are beneficial for readers not familiar with the Ifugao Rice Terraces. However, the resolution needs to be improved significantly. For instance, the legend on one photo is illegible. A figure/photo with a closeup of the terraces would also be of interest for readers not familiar with this heritage.

  • Addressed

Photographer?

  • Addressed

Ethnographic accounts and community stories

The information presented in this chapter is interesting. There is however not much information on method and methodological approaches on for instance documenting and utilizing indigenous knowledge in this context (Ifugao). Methodological and ethical consideration

  • We approached ethnographic accounts in Ifugao with a strong focus on cultural sensitivity and ethical considerations. We particularly underscored the respect and the amplification of the voices and perspectives of the Ifugao community while conducting research and interpreting ethnographic accounts. We understand that the Western-centric interpretation of the terraces had long romanticized and frozen the Ifugao in time, obscuring their dynamic history of resilience and adaptation. Ethical considerations guided every step of our research and actively engaged with the Ifugao community, forging partnerships that prioritized the community's interests and autonomy.

The paper mentions that one of the authors is an Ifugao but does not clarify why this is important to specifically mention, e.g. as a holder of knowledge, as having a perspective that allows for a better understanding or so on.

  • Martin, being an Ifugao, possesses a unique ability to contextualize oral historical accounts within the cultural nuances of the Ifugao community. This expertise allows him to convey insights that may be challenging for non-Ifugao individuals to discern.

Nature of archaeology: hypothesis testing

A link to the Ifugao Archaeological Project website might be useful for those interested in finding more information about the archaeological research and the sites.

  • Addressed - For detail on the work of the IAP, please visit ifugao-archaeological-project.org.

Archaeological data

OK presentation. Well enough referenced.

Regarding Figure 2
The caption, yellow highlights of the word axis

Archaeology and community

No explanation or reference for, for instance the muyung traditional forestry system

Parts of the text read as concluding remarks and statements. A summary/conclusion?

Changing the narrative, empowering descendant communities

Line 688: he? Should it be we instead? If not, who is he?
Same for line 694 (his).

  • addressed

This chapter runs through several important topics quite briefly, several by using examples not Ifugao. The examples are relevant, but the relevance in terms of Ifugao and the Ifugao Rice Terraces could perhaps be an even stronger focus in this part. The various topics/sub-chapters are brief, it might be beneficial to chose fewer topics and focus more on one or two questions.

In the final paragraph, it is briefly mentioned that community archaeology is not perfect and does not answer all problems. This is an important point, but at no point prior has there been any mention or discussion of the associated challenges or weaknesses.

  • Deleted all of the subsections to focus on the Ifugao example.

Regarding figure 3.

The resolution of especially one of the pictures could be better, but it is not quite as bad as figure 1.

  • Addressed – replaced with high resolution image

Concerning the bottom picture of grade five students where some faces are seen clearly enough to be identifiable. Have there been any consideration concerning ethical issues regarding this? Is there parental permission to publish this picture of minors? If so, this should be made clear in the caption. However, removing the picture of the children is an option that should definitely be considered.

  • Addressed – replaced with an image without the students

Photographer?

  • Addressed – added info

Reviewer 3 Report

Comments and Suggestions for Authors

This is a well written (a bit long) presentation that set about to address the origins of the terraces Ifugao in the context of decolonization.  For archaeologist, it is important to work with the local community and to have the research in the context of the local and when possible Indigenous knowledge. The landscapes archaeologists work in will have meanings to local peoples that will enhance the research and give significance to the settings. This is the strength of this article. 

There are issues however that are disconcerting. The authors are interested in data driven discussions, yet we are not afforded these. There is some repetition and it would benefit from consolidation. Also, the section on Nationalism seems out of place. What is the purported origin narrative of the terraces?  This might be easier to gather if the information is presented in a different order.  There is little doubt that the issues that are raised are not only important, and worth of publication, yet the manner in which it is divulged is problematic.

The first concern regards the age of the terraces.  There are only 2 citations that are referenced for the proponents of the attribute the age of 2,000 years for the Ifugao Rice Terraces, one is 1919 and the other is 1955.  When you consider that the 1919 was simple speculating int eh establishment of the Indigenous law, that is expectable at that time.  The second is by an individual who was truly and amateur who married a Ifugao to live and dedicate his life to the Philippines. You can find the UNESCO World Heritage testament, yet that must be based on some information.  While the introductions asserts that there are numerous academic articles as well as presentation in popular media; these are however not cited. It is hard to believe that such an entrenched narrative could have been settled based on 2 articles from 1919 and 1955?  The acceptance of the age of the terraces needs to be better grounded.

Another issue is the presentation organization.  It would seem best to set the controversy, brining in the Archaeological data to show the incongruence, and then develop the oral historical base that appears to match the actual archaeological data.  By reorganizing the presentation to validate the title that older is not better would highlight the dispute, and bring important confirmation to the local Indigenous history.  As it is now, we read so much of the local history without understanding the relevance.  The archaeological data are presented but not correlated with the local history. If the archaeological data were presented first and the local history second with dates/tables that would demonstrate the fit to excavation data, it would help to confirm the new position that is espoused. Decolonizing Archaeology might be better towards the end before Archaeology and Community?

The archaeological data, too, present some issues.  The tables are not properly referenced, Table 1 is 2 and 2 is 1.  Further who is Conklin on the title of Table 1?  Is the Harold Conkiln?  And is the reference of 27 to his work published in 1980?  Conklin is a remarkable individual and was so well respected by the Indigenous of the Philippines. Did he subscribe to the narrative too? How does Robert Maher's research relate to the current piece?  The dates vary form very early to late, and more that 1000 years and even greater than 1500 are not outside the possibilities.

Author Response

Reviewer 3

This is a well written (a bit long) presentation that set about to address the origins of the terraces Ifugao in the context of decolonization.  For archaeologist, it is important to work with the local community and to have the research in the context of the local and when possible Indigenous knowledge. The landscapes archaeologists work in will have meanings to local peoples that will enhance the research and give significance to the settings. This is the strength of this article. 

There are issues however that are disconcerting. The authors are interested in data driven discussions, yet we are not afforded these. There is some repetition and it would benefit from consolidation. Also, the section on Nationalism seems out of place. What is the purported origin narrative of the terraces?  This might be easier to gather if the information is presented in a different order.  There is little doubt that the issues that are raised are not only important, and worth of publication, yet the manner in which it is divulged is problematic.

  • This paper aims to demystify, and in the process, decolonize, archaeological and historical narratives developed in the early 1900s by pioneer anthropologists who worked in the Cordillera region of the Philippines. In addition, the dating of the Ifugao Rice Terraces has been part of the nationalist discussions that older is better, but at the same time, relegating the Ifugao in the sidelines of history. This can be achieved by actively engaging stakeholder and descendant communities, as exemplified by our work.

The first concern regards the age of the terraces.  There are only 2 citations that are referenced for the proponents of the attribute the age of 2,000 years for the Ifugao Rice Terraces, one is 1919 and the other is 1955.  When you consider that the 1919 was simple speculating int eh establishment of the Indigenous law, that is expectable at that time.  The second is by an individual who was truly and amateur who married a Ifugao to live and dedicate his life to the Philippines. You can find the UNESCO World Heritage testament, yet that must be based on some information.  While the introductions asserts that there are numerous academic articles as well as presentation in popular media; these are however not cited. It is hard to believe that such an entrenched narrative could have been settled based on 2 articles from 1919 and 1955?  The acceptance of the age of the terraces needs to be better grounded.

  • There are only two published materials on this assumed origin of the terraces. It has become entrenched because it was part of the Philippine history curricula that all Filipinos educated in the Philippines were/are exposed to.

We added this:

  • While only two published materials purports to argue for the 2,000 years origin of the terraces, this has become entrenched in the Philippine historical narratives because of the inclusion of the dating on Philippine curricula.

Another issue is the presentation organization.  It would seem best to set the controversy, brining in the Archaeological data to show the incongruence, and then develop the oral historical base that appears to match the actual archaeological data.  By reorganizing the presentation to validate the title that older is not better would highlight the dispute, and bring important confirmation to the local Indigenous history.  As it is now, we read so much of the local history without understanding the relevance.  The archaeological data are presented but not correlated with the local history. If the archaeological data were presented first and the local history second with dates/tables that would demonstrate the fit to excavation data, it would help to confirm the new position that is espoused. Decolonizing Archaeology might be better towards the end before Archaeology and Community?

  • We decided to forefront communities/oral historical narratives to stress the importance of community perspectives. Presenting archaeological data first reifies this seeming hierarchy of knowledge production.

The archaeological data, too, present some issues.  The tables are not properly referenced, Table 1 is 2 and 2 is 1.  Further who is Conklin on the title of Table 1?  Is the Harold Conkiln?  And is the reference of 27 to his work published in 1980?  Conklin is a remarkable individual and was so well respected by the Indigenous of the Philippines. Did he subscribe to the narrative too? How does Robert Maher's research relate to the current piece?  The dates vary form very early to late, and more that 1000 years and even greater than 1500 are not outside the possibilities.

  • Provided more details: Radiocarbon determinations from Banaue Ifugao, obtained by Maher [25, 28] and Harold Conklin [27]. While some of these radiocarbon determinations show older (>500 years), these samples are not contextualized and may not be associated with the archaeological event in question. For more discussion, refer to Acabado and colleagues discussions [3, 5].
  • Fixed the table numbering

Round 2

Reviewer 3 Report

Comments and Suggestions for Authors

This version of this paper is much clearer and yet there are still issues.

The point is that there has been an adoption of an assertion of longevity that the work of these scholars wish to cast doubt. They also wish to enlist the reader to value the local histories and integrate these into a fuller narrative.  Also, there is clear imperative of more inclusive research that decolonizes archaeology and engages local communities at the forefront. This is a strength of the paper and very important in the development of archaeology in the context of descendant communities.  The effort to “decenter” Western views and to give priority to Indigenous perspectives and showing respect is the most significant contribution. There needs to be clear links of the profound impact of the leading narrative and the importance of this new approach where archaeological evidence is paired with oral histories.

The need for a fuller narrative is well presented in broad brush yet one not steeped in Philippine history and in particular the nature of the dominant narrative of a 2,000-year chronology of terraces, there is much missing in the discussion.  Early anthropologists “posited that the Ifugao Rice were over 2,000 years old.”  Yet they suggest it is an assertion and presents the local group as unchanging.  This is hard to believe as so much is always changing and Dove’s obituary on Conklin suggests that his work there was at a propitious time of development and use, implying development and change. Terrace are constructed where water moves too fast and develop along with the growth of population, how does this fit the proposed narrative? And how does this concord with the practice and origin of muyung that is called traditional forestry.

Are there any documents that cast the terrace as passive relics? Examination of the UNESCO page, there is the use of the term dynamic living landscape.  It  does imply 2,000 years of  rice cultivation. That is one point that the authors suggest is not correct, but have the local groups lived there for that time?  Is the use of rice restricted to ponds? How might one dismiss rice cultivation? The terraces are well developed and it would seem that there is a developmental chronology for the terraces. And the areas are used for a number of annual and perennial resources.

Work is on one site with a proposed  ~500-year chronology, is this a basis for interpreting the whole area? Terraces are notoriously difficult to date. How do the ethnohistorical observations fit into the alternative narrative asserted by the authors? The reader is told that there were movements of peoples at the time of the Spanish Conquest. This movement was to escape the brutalities?  The investigated area was unoccupied at this time (hard be fathom)? What do colonial records indicate? Is there evidence of clashes of populations with the movements from one area to this area?  Is the reader to assume that taro farmers were not related to the local population today and the shift to rice was part of the movement of populations with the Spanish Invasion? This background is unclear.  One could go to all the authors personal references, yet this should stand on its own and enlist the reader to pursue the details. The dots need to be connected.

The archaeological tables, while readable, are a bit confusing. These span much of the past before and after Spanish.  There is a lot of data on materials and the archaeo botanicals and phytoliths  (no citations of Peardall 90 or Bodner 86?) yet these archaeological details supersedes the thrust of  the paper on Indigenous participation. A good summary of the point: dating and taro v rice island reference th backup reports? It is sharply in contrast to the rest of the paper.

  This paper would be stronger if it started with the idea that there is a controversy and that controversy can be resolved in partnership of archaeology and indigenous citizen scientists.  A background of the ethnohistory would be important.   Then to present the archaeological “narrative” that has been promulgated in the context of the contrasting archaeological results. This turns to the importance of engaging the local communities and working to address differences.  More on the valuable partnership as discussed to build to a conclusion that is given.   

Repetitive even in sections, care in editing is required.

How is it that any scholar and anthropologist would say something is unchanging? Especially terraces related to subsistence.

Map is not very helpful.  Yes the major sites are shown, but many other places are mentioned in the text as neighboring locales yet where? Tokak, Cambulo, Amduntog, Hapao, Nagacadan, Banaue among a few.  The map would be helped by showing roads and major cities and some characteristics other than monotonous green to orient the reader.  The lines might be political districts but not known why they are there.

Many statements of active landscape not static monuments repetition is good, but needs to be well placed.

Also the use of isolation

Comments on the Quality of English Language

English is good, here is repetitions that are an issue of editing. The draft still have organizational logical challenges in the presentation.

Author Response

I wanted to thank the 2nd round reviewer for some thoughtful comments and suggestions. I would like to address last point that they emphasize, “How is it that any scholar and anthropologist would say something is unchanging? Especially terraces related to subsistence.” This is a misreading of the manuscript, which is either a shortcoming of the writing style/presentation or something that was missed by the reviewer. In any event, we responded to this comment by emphasizing that the terraces are representations of a dynamic culture in the abstract:

Abstract: This study examines the intersection of archaeological data and community narratives in interpreting the Ifugao Rice Terraces in the Philippines, a UNESCO World Heritage Site. Long regarded as 2,000-year-old symbols of an uncolonized cultural past, recent research challenges this view, suggesting a 16th-century origin coinciding with Spanish contact. The longstanding characterization of the Ifugao Rice Terraces as 2,000-year-old monuments cemented a perception of Ifugao culture as static and unchanging, overshadowing the dynamic cultural practices that have persisted and evolved over the centuries. It is crucial to recognize that these terraces are not frozen in time but are active representations of Ifugao's living culture, which has continually adapted to social, environmental, and historical changes while maintaining its distinct identity. This paradigm shift, supported by radiocarbon dating and ethnohistorical analysis, aligns more closely with local oral histories and portrays the Ifugao not as passive inheritors of tradition but as active participants in their history. We argue for the integration of scientific data with community stories, presenting a holistic understanding of the terraces as dynamic elements of Ifugao resilience and identity. The findings advocate a move away from romanticized historical interpretations toward a narrative that respects the complexity and adaptability of indigenous cultural landscapes.

For our point-by-point response to this comment, here’s our response:

  • It is not the terraces, it is the narrative about the 2,000-year old model. Nothing in this manuscript suggests that the terraces are static, on the contrary, the manuscript is arguing that the terraces are very dynamic, but the old history dating erases that dynamism.
  • Added this line in the ms to stress the point: The terraces are reflective of cultural and historical dynamism of the Ifugao, but the 2,000-year old origin argument appear to make these terraces as static.

For the more substantive comments, the reviewer writes”

“The need for a fuller narrative is well presented in broad brush yet one not steeped in Philippine history and in particular the nature of the dominant narrative of a 2,000-year chronology of terraces, there is much missing in the discussion.  Early anthropologists “posited that the Ifugao Rice were over 2,000 years old.”  Yet they suggest it is an assertion and presents the local group as unchanging.”

Our response: 

  • Thank you for these comments. We added a series of subsections to make it clear that the pioneer anthropologists did not use any archaeological data to support their claim. And this claim has become the de facto truth because of how history curriculum in the Philippines adhered to the long history and isolationist models.

Here are the additional subsections:

1.2. Changing the Narrative

The 2000-year dating of the Ifugao Rice Terraces has played a pivotal role in shaping perceptions of Ifugao culture as static and unchanging through millennia. This interpretation, deeply rooted in the early anthropological work of scholars like Roy Franklin Barton [1] and Henry Otley Beyer [2], posit the terraces as ancient achievements of agricultural engineering, constructed over two millennia ago. The assertion of such antiquity has been intertwined with the narrative of an unchanging culture, one that has ostensibly remained isolated and untouched by the broader historical and cultural dynamics that have swept through the rest of the Philippines and Southeast Asia.

1.2.1. Implications of the 2000-Year Dating

The 2,000-year old construction dating assumes static culture since archaeological models strongly argue that if intensive agriculture has been documented, then we expect to see developments in other aspects of culture. It is then assumed that the Ifugao did not do anything else but plant rice for two millennia. This perspective also suggests that the Ifugao people have managed to preserve their way of life, with little to no external influence, maintaining the same techniques of rice cultivation and terrace construction passed down through generations. It romanticizes the Ifugao as custodians of an ancient tradition, living relics of a precolonial past. This narrative has been appealing because it offers a sense of timelessness and purity, resonating with nationalist sentiments that valorize indigenous cultures as symbols of national identity and resistance to colonial forces.

1.2.2. The Notion of Isolation

Closely related to the idea of an unchanging culture is the notion of isolation. The dating implies that the Ifugao Rice Terraces—and by extension, the Ifugao people—have existed in a bubble of isolation, shielded from the transformative influences of trade, colonialism, and cultural exchange that have characterized the history of the region. This perception of isolation has been used to support the argument that the terraces are a testament to indigenous ingenuity and resilience, constructed and maintained without external influence.

However, this view oversimplifies the complex historical and cultural dynamics that have shaped the Ifugao region. It overlooks evidence of trade, interaction, and conflict with neighboring groups and colonial powers, which have undoubtedly influenced Ifugao society. Moreover, it disregards the adaptability and agency of the Ifugao people, who have navigated the challenges posed by external pressures and internal developments over centuries.

1.2.3. Critique and Reevaluation

Recent archaeological findings and ethnohistorical research challenge the narrative of an unchanging, isolated culture. Studies suggesting a much later inception of the terraces, around the time of Spanish colonization in the 16th century, highlight the dynamic nature of Ifugao society. This evidence indicates that the terraces are not relics of a static past but rather the outcome of a society's adaptive strategies in response to social, environmental, and political pressures.

The reconsideration of the terraces' dating prompts a broader reflection on how cultures are perceived and represented in historical narratives. It underscores the need to move beyond simplistic notions of timelessness and isolation, recognizing the fluidity and interconnectedness of human societies. Acknowledging the terraces as a product of historical contingencies rather than a symbol of unchanging tradition allows for a more nuanced understanding of the Ifugao's cultural resilience and ingenuity.

The 2000-year dating of the Ifugao Rice Terraces and the associated ideas of an unchanging, isolated culture have significantly influenced perceptions of Ifugao society. However, emerging archaeological and ethnohistorical evidence challenges these notions, highlighting the dynamic and interconnected nature of Ifugao history. This reevaluation not only enriches our understanding of the Ifugao but also contributes to broader discussions about the representation and interpretation of indigenous cultures in historical narratives.

Another comment: “This is hard to believe as so much is always changing and Dove’s obituary on Conklin suggests that his work there was at a propitious time of development and use, implying development and change. Terrace are constructed where water moves too fast and develop along with the growth of population, how does this fit the proposed narrative? And how does this concord with the practice and origin of muyung that is called traditional forestry.”

Response: Adding a section discussing this will lengthen the already long manuscript. Readers can perhaps look at publications cited in the manuscript that discusses these issues. The comment, “(T)errace(s) are constructed where water moves too fast and develop along with the growth of population, how does this fit the proposed narrative,” is not accurate. Fast moving water will destroy the terraces, hence the kilometers-long irrigation channels to slow the flow of water. This is dealt with in another piece, cited in the manuscript.

The purpose of citing muyung is to emphasize the role of Indigenous knowledge in landscape management. This was contextualized in the ms: “The integration of archaeological findings and Indigenous narratives particularly shines in the Ifugao Rice Terraces' context. It reveals a dynamic history of sustainable adaptation to climatic challenges, with indigenous knowledge guiding effective water and soil management techniques. This indigenous wisdom, embedded in local stories and integrated with the landscape's history, lays a robust foundation for climate-resilient agricultural practices. Moreover, this collaboration extends to natural resource management, evident in the Ifugao's muyung traditional forestry system [15,16].

Comment #3: “Are there any documents that cast the terrace as passive relics? Examination of the UNESCO page, there is the use of the term dynamic living landscape.  It does imply 2,000 years of rice cultivation. That is one point that the authors suggest is not correct, but have the local groups lived there for that time?  Is the use of rice restricted to ponds? How might one dismiss rice cultivation? The terraces are well developed and it would seem that there is a developmental chronology for the terraces. And the areas are used for a number of annual and perennial resources.”

Our response:

  • The UNESCO convention and OUV categories are problematic. It is based on the enlightenment philosophy that tends to erase local realities. The UNESCO nomination dossier DOES NOT HAVE ANY archaeological data presented, only the speculation of Barton and Beyer. The state party nominating the sites could have cited Maher’s works, who argued that there is no support for the dating. At the end of the day, the dating is not important in the nomination as these are not listed in the Archaeological Sites category, they are listed in the Cultural Landscapes category.

Comment #4: “Work is on one site with a proposed ~500-year chronology, is this a basis for interpreting the whole area? Terraces are notoriously difficult to date. How do the ethnohistorical observations fit into the alternative narrative asserted by the authors? The reader is told that there were movements of peoples at the time of the Spanish Conquest. This movement was to escape the brutalities?  The investigated area was unoccupied at this time (hard be fathom)? What do colonial records indicate? Is there evidence of clashes of populations with the movements from one area to this area?  Is the reader to assume that taro farmers were not related to the local population today and the shift to rice was part of the movement of populations with the Spanish Invasion? This background is unclear.  One could go to all the authors personal references, yet this should stand on its own and enlist the reader to pursue the details. The dots need to be connected.”

Our response:

  • We tested 5 (five) sites, at least for the IAP. It was mentioned in the submissions: Contrary to the notion that the Ifugao rice terraces are 2,000 years old—an assumption unsupported by any archaeological evidence. Recent archaeological and ethnographic data from five major sites including Old Kiyyangan Village, Hapao, Nagacadan, Batad, and Banaue suggest a much shorter history. The absence of evidence should not be mistaken for evidence of absence; yet, the lack of archaeological support for the terraces' ancient origins cannot be ignored.

  • On top of these, we cited Maher’s work on multiple (7) sites (below), plus Bodner’s work in Bontoc.

  • On the question about taro, we added the work that shows that there were populations prior to rice growers. We wrote, “Paleoethnobotanical analysis indicated a potential presence of wet-rice cultivation dating back to 675 years before present (BP), but low counts of rice phytoliths suggested possible displacement within the soil column due to water seepage [36,37], casting doubt on earlier cultivation theories. Residue analysis from pottery did not indicate rice production, but instead showed evidence of taro consumption, further supporting the argument for the later emergence of wet-rice agriculture in Ifugao.”

Readers should be able to deduce that the dating for taro and the AMS dates in the OKV are earlier than the rice appearance, hence, there were people living in the region.

  • This dating methodology is contextualized in the hypothesis testing section, there is no way to test the whole universe. Besides, this work is based on five sites, including Maher’s and Conklin’s C14 dates, although majority is from the OKV site. Table below shows the AMS dates from Banaue:

Lab. No.

Unit

DBS

Layer

CRA

13C

Cal AD (BCal)

Post-AD 1585 Probability*

AA78973

Mamag

0.855

II

119±38

25.2

1687-1862

74.6%

AA78974

Mamag

1.3m

III

485±39

-27.5

1325-1460

74.6%

AA78971

Rasa

0.35m

II

313±38

-24.4

1620-1800

98.5%

AA78972

Rasa

.52m

III

164±38

-26.0

1527-1757

98.5%

AA78969

Linagbu

0.55m

II

180±38

-26.5

1736-1867

99.9%

AA78970

Linagbu

0.75m

III

131±38

-29.3

1663-1753

99.9%

AA78975

Achao

.075m

II

193±38

-25.0

1646-1809

N/A

Maher’s dates:

Site/Locality

Depth (cm)

Lab #

Material

14C BP

Cal. CE (2 σ – 95%) (recalibrated using IntCal13)

14C Sample context

Reported by

If1 - Nabyun

91

GX0668

M. sinensis

205 ± 100

1493-postbomb

Pond-field

[25]

If2 - Nabyun

91

GX1900

No data

325 ± 110

1408-postbomb

House Platform

[25]

If2 - Nabyun

90

GX1901

No data

695 ± 100

1052-1435

Midden

[25]

If2 - Nabyun

90

BX2184

No data

735 ± 105

1043-1413

Midden

[25]

If3 - Banaue

4.4

GX2183

No data

2950 ± 250

1867-540BCE

House Platform

[25]

Gawwa, Poitan

5

GX3138

No data

530 ± 140

1192-1792

Underground chamber

[25]

Gawwa, Poitan

5

GaK5238

No data

530 ± 100

1273-1631

Underground chamber

[25]

Lugu

No data

UGA2515

No data

395 ± 60

1430-1639

Terrace embankment post

[27]

If20 - Banghallan

50

GaK6442

No data

890 ± 310

434-1647

Village edge

[28]

If20 - Banghallan

60

UGA1541

No data

1340 ± 375

176BCE-1388CE

Village edge

[25]

Table 2. Radiocarbon determinations from Banaue Ifugao, obtained by Maher [25, 28] and Harold Conklin [27]. While some of these radiocarbon determinations show older (>500 years), these samples are not contextualized and may not be associated with the archaeological event in question. For more discussion, refer to Acabado and colleagues’ discussions [3, 5].

Comment #5: “The archaeological tables, while readable, are a bit confusing. These span much of the past before and after Spanish.  There is a lot of data on materials and the archaeo botanicals and phytoliths (no citations of Peardall 90 or Bodner 86?) yet these archaeological details supersedes the thrust of  the paper on Indigenous participation. A good summary of the point: dating and taro v rice island reference the backup reports? It is sharply in contrast to the rest of the paper.”

Bodner’s work is cited in the manuscript since V1:

  • In the neighboring region of Bontoc, Bodner's [40] work has provided compelling evidence for the later inception of wet-rice cultivation, postdating 1600 CE. This assertion is grounded in the notable absence of archaeobotanical data indicative of wet rice prior to the 1600s. Supporting this perspective, an accumulation of data from various scholarly pursuits—including Maher's investigations, Bodner's own dissertation, Conklin's landscape research, and the ongoing Ifugao Archaeological Project—converges to suggest a significant shift in agricultural practices to wet-rice cultivation occurring relatively recently.

Comment #6: “This paper would be stronger if it started with the idea that there is a controversy and that controversy can be resolved in partnership of archaeology and indigenous citizen scientists.  A background of the ethnohistory would be important.   Then to present the archaeological “narrative” that has been promulgated in the context of the contrasting archaeological results. This turns to the importance of engaging the local communities and working to address differences.  More on the valuable partnership as discussed to build to a conclusion that is given.”   

Response (added to the intro): We underscore the deeply ingrained controversy centered around the ethnohistory of the Ifugao Rice Terraces in the Philippines, a UNESCO World Heritage Site, and proposes a pathway to resolution that involves collaboration between archaeologists and the Ifugao community. It foregrounds the notion that the traditional archaeological narrative, heavily influenced by early 20th-century anthropologists like Roy Franklin Barton [1] and Henry Otley Beyer [2], has promulgated a view of the Ifugao and their terraces as static relics of a distant past. This perspective, deeply entangled with nationalist pride, asserts the terraces' antiquity as a symbol of uncolonized purity and cultural identity.

Comment #7: Map is not very helpful.  Yes the major sites are shown, but many other places are mentioned in the text as neighboring locales yet where? Tokak, Cambulo, Amduntog, Hapao, Nagacadan, Banaue among a few.  The map would be helped by showing roads and major cities and some characteristics other than monotonous green to orient the reader.  The lines might be political districts but not known why they are there.

Reponse: Map updated and added information on the caption: Figure 1. Ifugao Province, with approximate locations of major sites mentioned in the text (Map by M. Echavarri and the IAP). The municipality of Asipulo includes Tokak and Amduntog; Cambulo is adjacent to Batad; and, Old Kiyyangan Village is adjacent to Nagacadan.